

# Combining data from different sampling methods to study the development of an alien crab *Chionoecetes opilio* invasion in the remote and pristine Arctic Kara Sea

Anna K. Zalota[1], Olga L. Zimina[2] and Vassily A. Spiridonov[1]

[1] Shirshov Institute of Oceanology, Russian Academy of Sciences (SIO RAS), Moscow, Russia
[2] Murmansk Marine Biological Institute KSC, Russian Academy of Sciences (MMBI KSC RAS), Murmansk, Russia

## ABSTRACT

Data obtained using three different types of sampling gear is compared and combined to assess the size composition and density of a non-indigenous snow crab population *Chionoecetes opilio* in the previously free of alien species Kara Sea benthos. The Sigsbee trawl has small mesh and catches even recently settled crabs. The large bottom trawl is able to catch large crabs, but does not retain younger crabs, due to its large mesh. Video sampling allows the observation of larger crabs, although some smaller crabs can also be spotted. The combined use of such gear could provide full scope data of the existing size groups in a population. The density of the crabs was calculated from the video footage. The highest figures were in Blagopoluchiya Bay at 0.87 crabs/m$^2$, where the settlement seems to be reaching its first peak of population growth after the introduction. High density in the Kara Gates Strait at 0.55 crabs/m$^2$, could be due to the close proximity of the Barents Sea from where the crabs can enter by both larval dispersal and active adult migration. All size groups have been present in most sampled areas, which suggest successful settlement and growth of crabs over a number of years. Again, this was not the case in Blagopoluchiya Bay with high density of small crabs (<30 mm CW), which confirms its recent population growth. Male to female ratio was strikingly different between the bays of the Novaya Zemlya Archipelago and west of the Yamal Peninsula (0.8 and 3.8 respectively). Seventy five ovigerous females were caught in 2016, which confirms the presence of a reproducing population in the Kara Sea. The spatial structure of the snow crab population in the Kara Sea is still in the process of formation. The presented data indicates that this process may lead to a complex system, which is based on local recruitment and transport of larvae from the Barents Sea and across the western Kara shelf; formation of nursery grounds; active migration of adults and their concentration in the areas of the shelf with appropriate feeding conditions.

Corresponding author
Anna K. Zalota, azalota@gmail.com

## INTRODUCTION

The process of a non-indigenous species (NIS) invasion can take a long time (*Byers et al., 2015*) and often remains incompletely observed. An ongoing invasion of a large predatory

snow crab, *Chionoecetes opilio,* into the previously pristine Kara Sea benthic environment is an exceptional opportunity for researchers as as an invasion which has been both rapid and observable from the beginning of the process (*Zalota, 2017*). Due to the changing climate, and the current reduction of sea ice cover in the Arctic, it is widely expected that shipping and other human activity will greatly increase (*Ho, 2010*; *Liua & Kronbak, 2010*). Seas that are as productive and important for fishing as the Bering and the Barents Seas are well studied and have ongoing large scientific projects to study their populations of snow crabs, such as joint Russian–Norwegian studies of the Barents Sea (*Pavlov & Sundet, 2011*; *Hammer & Hoel, 2012*; *Jørgensen et al., 2015*; *Sokolov et al., 2016*). This is not the case for the Siberian seas that have lower productivity, and until recently had very short ice-free seasons (*Zenkevich, 1963*; *Vinogradov et al., 2000*; *AARI, 2009*; *Demidov & Mosharov, 2015*). Biological research in the Kara Sea benthos is mostly conducted by occasional expeditions of the Shirshov Institute of Oceanology, Russian Academy of Sciences (SIO), *Murmansk* Marine Biological Institute, Russian Academy of Sciences (MMBI) and the Knipovich Polar Institute of Fishery and Oceanography (PINRO) (*Jørgensen et al., 1999*; *Sokolov et al., 2016*). The intensity and area of sampling effort varies, and so does the sampling gear employed. Therefore, it is important to compare and understand the differences in the results obtained by different gear to be able to gather informative data on the new snow crab population.

The snow crab, *Chionoecetes opilio* (Decapoda: Oregonidae) invaded vast areas of the Barents and Kara seas with unprecedented speed for a shelf species (*Pavlov, 2006*; *Pavlov & Sundet, 2011*; *Zimina, 2014*; *Bakanev, 2015*; *Sokolov et al., 2016*; *Spiridonov & Zalota, 2017*; *Zalota, Spiridonov & Vedenin, 2018*). The native range of this species covers the North-Western Atlantic (Newfoundland and Labrador waters, south-west Greenland shelf to southern Baffin Bay) (*Squires, 1990*); the North Pacific northwards of the Aleutian Islands and the Sea of Japan (*Slizkin, 1982*), and the Chukchi Sea westward to the boundary with the East Siberian Sea and eastward to the Beaufort Sea (*Slizkin, Fedotov & Khen, 2007*; *Sirenko & Vassilenko, 2008*). There is only one record of snow crabs on the border between the East Siberian and the Laptev Seas, off the New Siberian Islands (*Sokolov, Petryashov & Vassilenko, 2009*). *Ch. opilio* is an active benthic predator consuming a broad range of invertebrates, and even fish (*Tarverdieva, 1981*; *Chuchukalo et al., 2011*; *Lovvorn, 2010*; *Kolts et al., 2013*; *Zalota, 2017*; *Zakharov et al., 2018*).

The first record of a snow crab in the Barents Sea was in 1996 (*Kuzmin, Akhtarin & Menis, 1998*). It is possible that the introduction took place between the mid-1980s and 1993 (*Alvsvåg, Agnalt & Jørstad, 2009*; *Strelkova, 2016*). By the mid-2010s the snow crabs occupied the entire central, eastern, and most of the northern part of the Barents Sea. Uncontrolled snow crab fishery commenced in 2013 in the international fishery enclave between the EEZs of Russia and Norway and in the Spitsbergen fishery protection zone (*Bakanev, Pavlov & Goryanina, 2017*; *Sundet & Bakanev, 2014*). A regulated snow crab fishery in Russia's EEZ of the Barents Sea began in 2016 (*Bakanev et al., 2016*).

The snow crab population grew in the Barents Sea and expanded towards the Kara Sea. The first crabs were found on the boundary of the two seas in 2008 (*Strelkova, 2016*), then in the north-west of the Kara Sea in 2010 and 2011 (*Strelkova, 2016*; *Zalota, Spiridonov &*

*Vedenin, 2018*). Both adults and larvae were caught in the south-western Kara Sea in 2012 (*Zimina, 2014*). In less than five years after the initial records, *Ch. opilio* was observed over the entire western Kara Sea shelf (*Zalota, Spiridonov & Vedenin, 2018*). A high abundance of adult snow crabs was recorded in 2013 in the south-western Kara Sea, between the Yamal Peninsula and the Kara Gate Strait, which is the entrance from the Barents Sea (*Strelkova, 2016*). In 2014, several size groups of juveniles were present throughout the western shelf and the fjords of the eastern Novaya Zemlya Archipelago, with the most numerous groups presumably originating from larval settling in 2013 (*Zalota, Spiridonov & Vedenin, 2018*). It is still uncertain if the Kara snow crab population is fully established and independent of the import of larvae and adult migration from the Barents Sea, and how far it can expand eastwards.

The oceanographic conditions of the Kara Sea are very different from the Barents Sea. The western Kara Sea is strongly influenced by the water exchange with the Barents Sea and by the advection of fresh water from large Siberian rivers' runoff (*Pavlov & Pfirman, 1995*; *Zatsepin et al., 2010a*; *Zatsepin et al., 2010b*; *Zatsepin et al., 2015*; *Polukhin & Zagretdinova, 2016*). The Kara Sea is covered with ice for most of the year, with extensive fast ice massifs and regular polynya formations (*Gavrilo & Popov, 2011*; *Polukhin & Zagretdinova, 2016*). Since the mid-2000s, the Kara Sea has followed the general Arctic trend of delaying sea ice formation in autumn and earlier decay in spring/early summer (*Ashik et al., 2014*). This coincided with the commencement of *Ch. opilio* invasion from the Barents Sea (*Zalota, Spiridonov & Vedenin, 2018*).

In comparison to the Barents Sea, the Kara Sea has a much lower primary productivity (*Vinogradov et al., 2000*; *Romankevich & Vetrov, 2001*; *Demidov & Mosharov, 2015*; *Demidov, Mosharov & Makkaveev, 2015*) and benthic biomass (*Zenkevich, 1963*; *Denisenko, Rachor & Denisenko, 2003*; *Kulakov et al., 2004*; *Udalov, Vedenin & Simakov, 2016*; *Chava et al., 2017*). Its ecosystem is noticeably affected by climate change and the lengthening of the ice-free season (*Ashik et al., 2014*). Persistent accumulation of organic pollution (*AMAP Assessment, 2015*), massive offshore and coastal oil and gas development, and shipping (*Amiragyan, 2017*) will impact the Kara Sea in the near future. The establishment of a breeding snow crab population, even if it remains dependent on the Barents Sea stock, may have an additional large-scale impact on the distinct Kara Sea ecosystem. On the other hand, the snow crabs could potentially grow to commercial size and become the target of regulated offshore fishery, which has not previously existed in the Kara Sea before. It is therefore critical to study development of the snow crab population in the Kara Sea to forecast the future of the Siberian Shelf ecosystems, and options for resource management and biodiversity conservation.

The Barents Sea snow crab population is spatially and temporally monitored in a standardized way (*Jørgensen et al., 2015*; *Strelkova, 2016*). Due to its limited fishery resources, the Kara Sea is less frequently visited and surveyed using standard fishery trawls, which makes it very difficult to obtain a representative samples of adult snow crabs and to monitor their abundance (*Zimina et al., 2015*; *Sokolov et al., 2016*). Since 2007, smaller scientific gear, such as Sigsbee trawls, has been used in regular expeditions of the SIO to the Kara Sea. It provides a good representation of juvenile groups, but most likely

underestimates large crabs (*Zalota, Spiridonov & Vedenin, 2018*). Therefore, to study the population of snow crabs in the Kara and potentially in other Siberian seas in the long term, we need to learn how to combine the data supplied by different gear to draw meaningful conclusions and obtain data any time there is an opportunity.

In the summer season of 2016 we employed three methods to study snow crab size composition and abundance: Sigsbee trawl, video transects; and a large Campelen-type bottom trawl. The video survey does not damage the seabed, and is a less costly and less labor intensive method of rapid assessment of the density and size structure of crabs' settlements. However, the video data lacks important information, such as differences in size composition related to sex ratio. Therefore, it is important to identify the information that can be combined safely.

The purpose of the present paper is to compare and understand the differences in the results obtained by different sampling gear to study an ongoing invasion of an alien crab in the remote Kara Sea. The data from three types of sampling gear is analyzed to obtain the size and sex composition, and the density of snow crab settlements in the Kara Sea in 2016. By identifying specifics and merging the results obtained by these different gear we aim to assess the progress of the *Ch. opilio* invasion in the Kara Sea, and to compare it with earlier stages (2008–2014) described by *Zimina (2014)*, *Strelkova (2016)* and *Zalota, Spiridonov & Vedenin (2018)*.

## MATERIAL AND METHODS

The crabs *Chionoecetes opilio* were studied using three sampling methods, during the cruises of the Research Vessel *Dalniye Zelentsy* (MMBI) and the RV *Akademik Mstislav Keldysh* (SIO), in August–September 2016. MMBI samples were collected using a Campelen-type bottom trawl with a 20 m horizontal by an 8–10 m vertical opening, equipped with a double net; the outer net with 135 mm mesh and lower insertion of the net with 12 mm mesh. The SIO samples were collected using a Sigsbee trawl with a steel frame of two m breadth and 35 cm height. The trawl was equipped with a double net; the outer net had 45 mm mesh and the inner net had 4 mm mesh.

A video transect was filmed by the SIO team of engineering and technical research in combination with four trawling samples (Fig. 1). This was done using an uninhabited, towed, submerged, inert vehicle (UTSI) *Video Module* (*Pronin, 2017*), equipped with a control and data transmission system through which information is received and control commands are transferred via an optical cable in real time. The UTSI *Video Module* has a navigation system, power supply, three video cameras (one of which is high resolution, set up to carry out planimetric surveys), six floodlight projectors and two laser scale indicators, with a set distance between them of 60 cm. The use of the UTSI *Video Module* allowed us to obtain geo-referenced (including depth), spatially oriented and scaled images of the bottom with organisms.

The team on the RV *Dalniye Zelentsy* collected crabs from 53 stations (further referred to as MMBI samples) on the west side of the Yamal Peninsula in the Kara Sea (Fig. 1, circles.). The RV *Akademik Mstislav Keldysh* collected trawl samples (further referred to
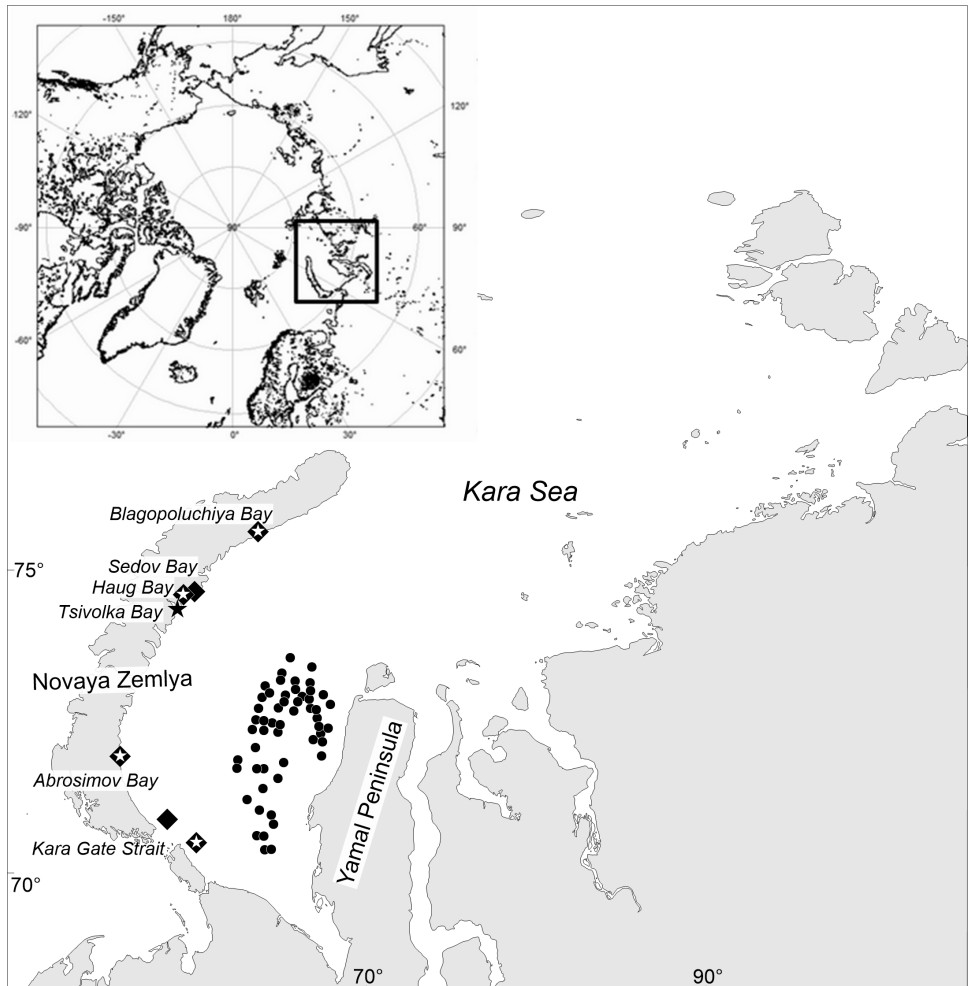

**Figure 1** Map of stations surveyed by the RV *Dalniye Zelentsy* (MMBI) and the RV *Akademik Mstislav Keldysh* (SIO RAS) in August–September 2016 in the Kara Sea. CIRCLES - MMBI bottom trawling stations; DIAMONDS - SIO RAS Sigsbee trawling stations; and STARS –stations with video footage of the bottom using UTSI *Video module*. (Maps created using PanMap; *Grobe, Diepenbroek & Siems, 2003*).

as SIO samples) in the vicinity of four Novaya Zemlya Archipelago bays and two in the area of the Kara Gates Strait (Fig. 1, diamonds). The video transects (further referred as video samples) were done prior to trawling at four of these stations (Fig. 1, stars in diamonds), and in the vicinity of Tsivolka Bay without a trawling sample (Fig. 1, black star). The trawling stations of SIO closely followed the route of the video transects and can therefore be directly compared. Both, video and trawling sampling was done in the vicinity of Blagopoluchiya, Haug, and Abrosimov bays, as well as in the southern region of the Kara Gates Strait (further referred to as Gates 2) (Fig. 1).

All crabs caught in trawls were sexed, based on visual characteristics, and measured (carapace width, CW) using a calliper to the nearest millimeter onboard the vessels. In this study crabs with CW less than 11 mm, whose sex cannot be easily identified by visual

inspection, are referred to as the "juveniles". The videos were viewed using the Media Player Classic—Home Cinema program in full screen mode. Manually, by screenshots, the videos were divided into still frames according to the changes in the bottom features to account for varying speed of towing, changing depth, and magnification. The height and width of the frame, the distance between two laser points and the carapace width (CW) of crabs present on the image were measured using a ruler. All frame measurements were converted to actual dimensions considering that the distance between the laser points on the bottom was 60 cm.

No native crabs, *Hyas araneus*, were spotted in these videos. However, in videos taken in other years (not discussed in this work) native crabs could be successfully differentiated from the snow crabs opilio in the vicinity of the Kara Gate Strait. Each video frame was visually inspected by one person (A. Zalota). This was done to minimize the errors related to different interpretations by various observers, which are unavoidable in such a subjective analysis. The frames where the viewer could not identify organisms easily were not used. This happened when the video module was raised too high over the sea floor (due to waves), but subject to visibility. Not all measured crabs were used in the density calculations, since some frames had to be cut around the edges to standardize the method of area calculations (in cases where the camera zoomed out and a circular lens was visible on the frame).

Most of the statistical calculations and analyses were performed using RStudio (*RStudio Team, 2016*; *R Core Team, 2018*). The size structure of collected crabs was analyzed using mixture model analysis in PAST software (*Hammer, 2013*). The best fit models were selected using the Akaike (*Akaike, 1974*) and log likelihood criteria. To analyze possible trends of crabs' size distribution in space, we looked at the correlation between the depth of sampling stations and different statistical parameters of the CW. Correlations were calculated using Microsoft Excel package.

## RESULTS

Overall, data was collected from 64 sample stations. MMBI trawled at 53 stations and caught 662 crabs; SIO caught 857 from 6 trawling stations and 884 crabs were caught on camera at the 5 stations where *Video Module* was employed (Fig. 1). In total 2,402 crabs were measured, which includes 1520 crabs caught in trawls.

### Size composition revealed by different sampling methods

Different methods of collection yielded diverse size distribution of crabs. Although the size composition of adult male and female snow crabs usually differs, we discuss their aggregate composition in order to compare the data from trawling with the video data, for which no sex differentiation is possible. The carapace width (CW) of crabs caught during SIO trawling ranged from 4 to 117 mm (Fig. 2A). Mixture analysis of CWs identified 9 distinct size groups from the bulk of SIO crabs (7 groups from the analysis and 2 were added manually to decrease noise during the analysis) (Table 1). The majority of crabs were of small size, with CW mode at 14 mm (while the mean was 16 mm) and another abundant group at CW 10 mm.
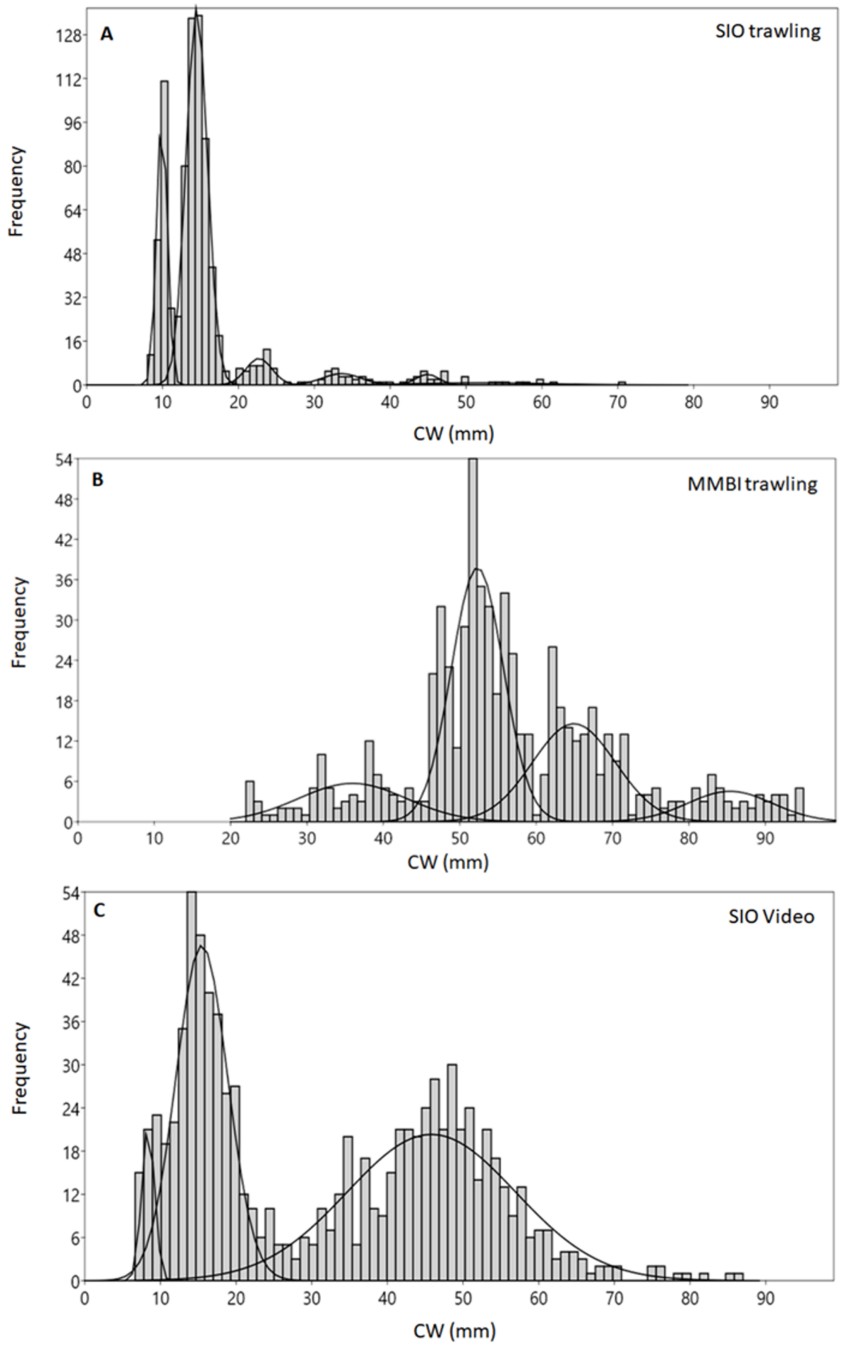

**Figure 2   Carapace width (CW) size group frequencies of *Chionoecetes opilio* collected and measured by different methods from the Kara Sea, 2016.** (A) All SIO samples collected by Sigsbee trawl; (B) MMBI samples collected by a large bottom trawl; (C) All data obtained from video footage.

The mixture analysis of crabs caught by MMBI trawling resulted in only 5 size groups (1 of which was added manually) (Table 1). Overall, crabs caught by this method were of larger size, with the minimum carapace width of 22 mm and maximum of 120 mm. A

large portion of crabs was within the 52 mm size group (mean was 57 mm) (Fig. 2B). The analysis merged smaller crabs into one group with large standard deviation (36 ± 7 mm). However, the mixture analysis identified more distinct large size groups (over 52 mm) in MMBI samples, than from SIO trawling and video sampling.

The smallest crab identified on the video had seven mm CW and the largest 127 mm. Mixture analysis yielded 4 size groups for this method (Table 1). Most frequently crabs were within two broad size groups with CW 15 ± 3 mm and 46 ± 11 mm (Fig. 2C). The mode CW was 15 mm, however for this method the mean CW differed at 32 mm.

The boxes in Fig. 2 (interquartile range containing 50% of the data) of Blagopoluchiya Bay are very similar for both methods, and include CW below 20 mm. The mode, median and means for the trawling and video samplings in Blagopoluchiya Bay are also very similar (Table 2). However, the maximum size observed in video (50 mm) was larger than in the Sigsbee trawling sample (38 mm), while the minimum was very similar (7 and eight mm respectively). Both methods yielded large number of crabs: 735 by trawling and 388 caught on video. The SIO trawling sample revealed more distinct and sharp size groups than the video sample (Fig. 3). However, the mixture analysis identified only one additional size group in the SIO trawling sample (Table 1).

The two methods had similar results for the south of the Kara Gates Strait (Gates 2, Fig. 4; Table 2). The central 50% of crabs had CW between 43 and 57 mm. Trawl sampling collected 20 crabs, and we observed 186 crabs on the video. While the minimum CW of crabs from trawling and video were very similar, 27 and 22 mm respectively, the video detected much larger crabs (127 mm) than the trawling (71 mm). However, we caught large crabs (up to 117 mm) while trawling in the nearby location (Gates 1, 2, Figs. 1, 4 Table 2).

We detected a very low number of crabs in Haug Bay using both methods (4 and 7 respectively). However, their sizes were very different. In trawl sample all crabs were 15–16 mm; in the video sample the sizes ranged from 27 to 50 mm, while the majority was within 42–47 mm.

The most distinct difference was observed in Abrosimov Bay (Figs. 4A, 4B Table 2). A large proportion of the trawl catch were small crabs (minimum and mode at four mm CW), and the central half of crabs were between 4 and 32 mm. On the video we could observe only larger crabs (minimum nine mm), and the central half of the crabs were much larger, between 34 and 48 mm. The largest crabs in both methods were very similar (50 mm trawling, 58 video). We measured twice as many crabs (88) on the video than in the trawling sample (43).

Sedov and Tsivolka Bays were sampled using different methods and cannot be compared directly. Trawling sample in Sedov Bay brought 28 crabs with 45 mm maximum and 10 mm minimum CW. On the Tsivolka Bay video we identified 215 crabs from 9 to 58 mm CW.

At the 53 MMBI trawling stations the minimum CW ranged from 22 to 72, and the maximum from 39 to 120. The modes and means ranged from 32 to 94 and 34 to 75 respectively. Most often, the minimum CW was 47 mm, maximum was 57 mm; mode and

Zalota et al. (2019), *PeerJ*, DOI 10.7717/peerj.7952

Peer J

**Table 1 Results of mixing model analysis of all *Chionoecetes opilio* carapace width (CW) measured from trawling (MMBI and SIO) collections and video footage from the Kara Sea in 2016.**

| Size groups/Instars | | I–II | III–IV | V | V | VI | VII | VIII | IX | X | >X | >X | Akaike IC | Log lk.hood |
|---|---|---|---|---|---|---|---|---|---|---|---|---|---|---|
| | | | | | | | **Mean CW ± standard deviation (mm)** | | | | | | | |
| All SIO | | 4* | 10 ± 1 | 14 ± 1 | 16 ± 1 | 23 ± 2 | 34 ± 3 | 45 ± 2 | 52 ± 9 | | | 100* | 1,743 | −866** |
| | | | | | | | | | | | | | 567 | −275*** |
| All MMBI | | | | | | | 36 ± 7 | | 52 ± 3 | 65 ± 5 | 85 ± 6 | 100* | 3,957 | −1,970 |
| All Video | | | 8 ± 1 | 15 ± 3 | | | | | 46 ± 11 | | | | 100* | 5,406 | −2,697 |
| Only from Blag-chiya Bay | SIO | | 10 ± 1 | 14 ± 1 | | 22 ± 1 | 34 ± 2 | | | | | | | 2,166 | −1,050 |
| | Video | | 8 ± 1 | 15 ± 4 | | | 32 ± 8 | | | | | | | 1,722 | −855 |
| All SIO | Female | | | <19* | | 22 ± 2 | 32 ± 3 | 45 ± 2 | 55 ± 2 | | | | 324 | −153 |
| All SIO | Male | | | <19* | | 24 ± 2 | 34 ± 4 | 45 ± 3 | | 63 ± 5 | | 114 ± 3 | 269 | −121 |
| All MMBI | Female | | | | | | 34 ± 7 | | 50 ± 3 | 60 ± 4 | 70 ± 2 | | 820 | −401 |
| All MMBI | Male | | | | | 24 ± 1 | 38 ±6 | 48 ± 1 | 53 ± 3 | 65 ± 2 | 79 ± 10 | 116 ± 3 | 3,153 | −1,562 |

**Table 2** General statistics of the carapace width (CW) distribution of crabs caught at SIO trawling (A) and video sampling (B) stations.

| A | | | | | | |
|---|---|---|---|---|---|---|
| **SIO trawling** | **Blagopoluchiya** | **Haug** | **Sedov** | **Abrosimov** | **Gates 1** | **Gates 2** |
| n | 735 | 4 | 28 | 43 | 27 | 20 |
| min-max | 8–38 | 15–16 | 10–45 | 4–50 | 10–117 | 27–71 |
| central 50% | 11–15 | 15–16 | 14–21 | 4–33 | 24–45 | 44–54 |
| mode/median/mean | 14/14/14 | 16/16/16 | 25/16/19 | 4/16/21 | 25/35/39 | 47/47/48 |

| B | | | | | |
|---|---|---|---|---|---|
| **SIO video** | **Blagopoluchiya** | **Haug** | **Tsivolka** | **Abrosimov** | **Gates 2** |
| n | 388 | 7 | 215 | 88 | 186 |
| min-max | 7–50 | 27–50 | 7–69 | 9–58 | 22–127 |
| central 50% | 12–19 | 38–47 | 34–52 | 34–48 | 43–57 |
| mode/median/mean | 15/15/17 | 47/47/42 | 50/46/41 | 48/43/40 | 43/48/51 |

**Notes.**

n, number of crabs measured; min-mas, minimum and maximum CW; central 50%, reflect the data between first and third quartile (50%) of CW distribution (see Fig. 4).

mean were 52 mm. Between 2 and 61 crabs were caught at the MMBI sampling stations, most commonly 3 crabs per station, but 12 crabs on average.

At the 53 MMBI trawling stations, the maximum, mode and mean CW sizes only weakly correlated with the depth (Table 3). In both SIO trawling and video samples the maximum CW sizes had strong correlation with the depth, although the sample size was very small.

When we mapped the maximum CW size for each station, we observed a trend in the MMBI samples, where larger crabs tend to be found in the south and northwards along the Yamal Peninsula. Towards the center of the western Kara Sea the maximum CW of the crabs decreases (Fig. 5). Similar mapping of ovigerous female findings did not show any observable trends.

## Sex ratio and sex related differences in size composition

Trawling allowed identifying and comparing crabs of different sexes. The male to female ratios were strikingly different between SIO (bays of the Novaya Zemlya Archipelago) and MMBI (west of the Yamal Peninsula) (Fig. 1): 0.8 and 3.8 respectively. There were only 8 ovigerous and 366 non ovigerous females in the SIO samples (ratio 0.02), while there were 72 ovigerous and only 67 non ovigerous females in the MMBI samples (ratio 1.07).

The central half of the CW size distribution of crabs differed between samples for all sexes, except for ovigerous females (Fig. 6). Ovigerous females were within a narrow size range of 44 and 58 mm in the SIO samples, and between 42 and 72 in the MMBI samples (Table 4). The non ovigerous females from SIO trawling samples ranged from 11 to 47 mm. The MMBI non ovigerous females were larger and ranged from 22 to 63 mm. Overall, male sizes also differed between these two sampling methods and area of collection. The CW of 293 crabs caught by SIO Sigsbee trawl ranged from 11 to 117 mm, and of the 523 crabs caught by MMBI large trawl ranged from 23 to 120 mm. Small mesh in the Sigsbee trawl allowed us to collect 190 juvenile crabs (less than 11 mm).

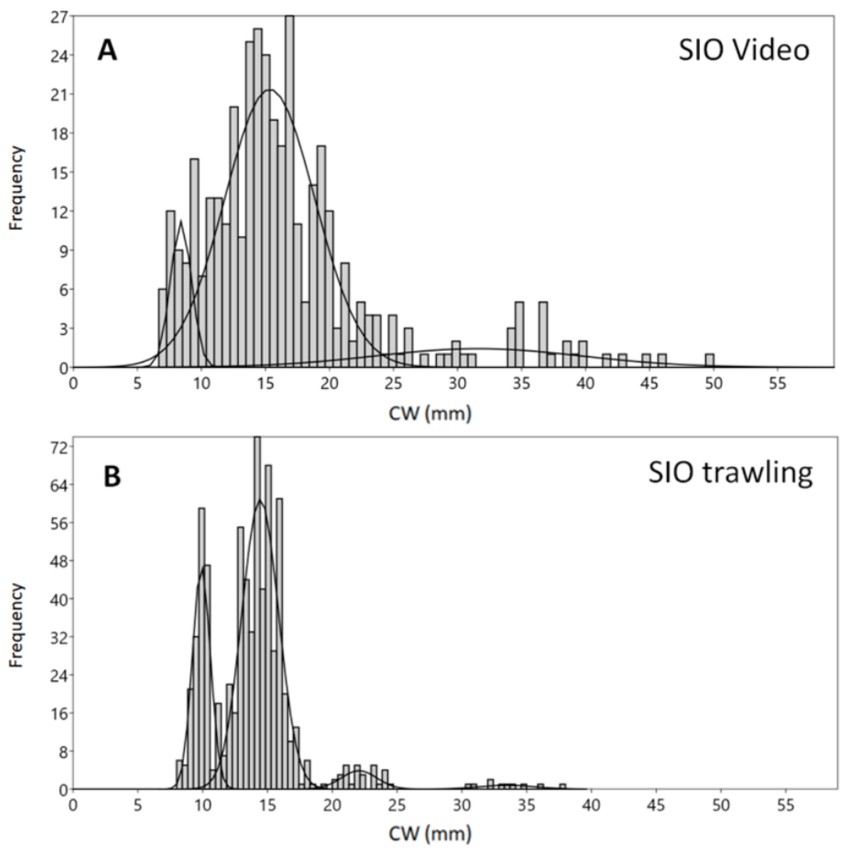

**Figure 3   Carapace width (CW) size group frequencies of *Chionoecetes opilio* collected and measured by different methods from the vicinity of Blagopoluchiya Bay in 2016.** (A) Video data collected prior to trawling; (B) SIO Sigsbee trawling.

## Abundance estimation

Population density of crabs was calculated from the video transects (Table 5). Overall 3,132 frames were analyzed, resulting in 3,776 m² of bottom inspected. The maximum density was observed near the Kara Gate Strait and in Blagopoluchiya Bay (0.87 and 0.55 crabs/m² respectively). In the other bays (located between the Kara Gate Strait and Blagopoluchiya Bay) the population density of crabs was several times or an order of magnitude lower, reaching the minimum value of 0.01 crabs/m² in Haug Bay (Table 5).

## DISCUSSION

### Advantages and disadvantages of the applied methods to study the snow crab population in the Kara Sea

The three methods discussed here revealed different aspects of the *Chionoecetes opilio* population size structure in the Kara Sea. The Sigsbee trawl used by SIO has small mesh and catches crabs as small as four mm CW, which is the size of recently settled crabs (*Conan et al., 1996*). However, it also has a small opening, and some large and agile crabs can escape. The video recording of the same area shows that large crabs are present,

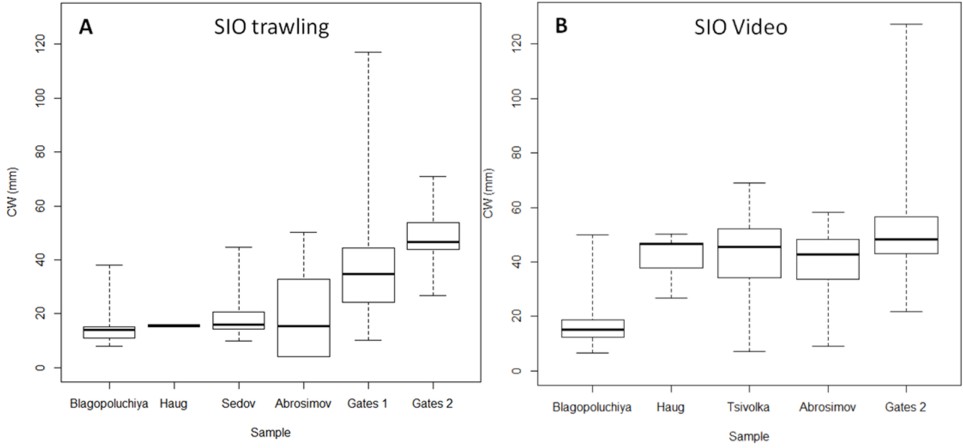

**Figure 4** **Box plots of carapace width (CW) distribution of crabs caught at SIO RAS stations.** (A) Trawling stations; (B) Video sampling stations. Boxes reflect data between first and third quartile (50%), thick line is the median, and whiskers extend to the maximum and minimum CWs (see Table 2).

**Table 3** **Correlation of carapace width parameters with the sample stations' depth of SIO RAS trawling (6 stations) and video (5 stations), and MMBI (53) stations.**

|  |  | Minimum | Maximum | Mode | Mean |
|---|---|---|---|---|---|
| MMBI | p | 0.004 | 0.48 | 0.54 | 0.37 |
| | $R^2$ | $2.0E^{-05}$ | 0.24 | 0.29 | 0.14 |
| SIO | p | 0.36 | 0.93 | 0.56 | 0.86 |
| | $R^2$ | 0.13 | 0.87 | 0.31 | 0.73 |
| Video | p | 0.53 | 0.91 | 0.07 | 0.53 |
| | $R^2$ | 0.28 | 0.83 | 0.004 | 0.28 |

although not always caught. The large bottom trawl is able to catch large crabs (22–120 mm CW), but does not retain younger crabs, due to its large mesh. We do however know that at least at some of the MMBI stations juvenile crabs were present. In some cases a similar to the SIO Sigsbee trawl was used, but the data was not dully recorded, and is thus omitted from the results. The combined use of this trawling gear could provide the full picture of the existing size groups in a population.

It is easy to observe larger crabs on the video, although some smaller crabs can also be spotted (up to seven mm CW) (Fig. 7A). The *Video Module* floats over the bottom with very little impact. Due to the muddy sediments in the studied area, every sudden movement of large agile organisms (crabs, fish) creates a cloud and can easily be spotted on the video. In all of the recorded footage, there were very few cases of such clouds: in most of them it was a fish, and sometimes crabs would run forward, and stop, therefore still recorded by us (Fig. 7C). It is safe to say that larger crabs (CW 30 mm and above) are quantitatively recorded on the video. However, crabs smaller than approximately 30 mm are probably substantially underestimated. Snow crabs are known to borrow in the sediments, especially in younger stages (*Conan et al., 1996*; *Dionne et al., 2003*). In some

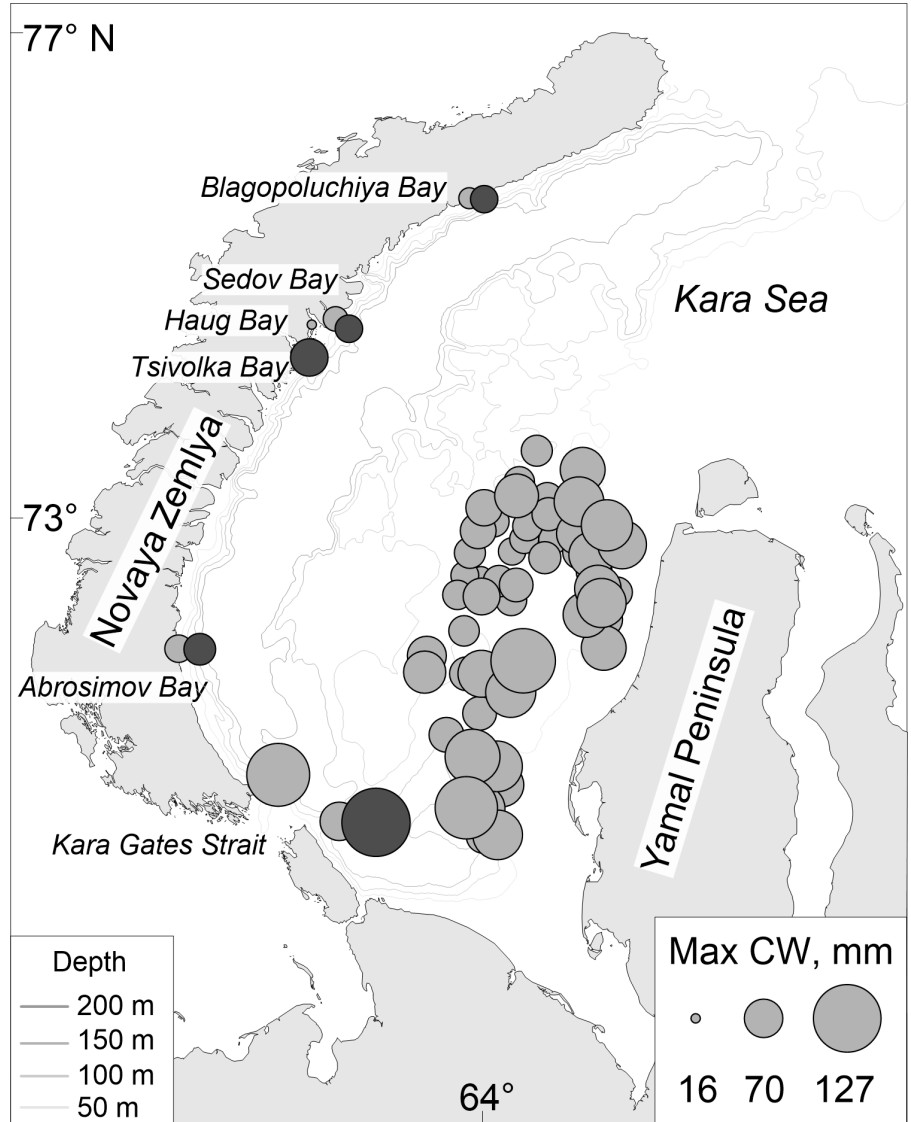

**Figure 5  Map of maximum carapace width of *Chionoecetes opilio* distribution collected using (light grey circles) MMBI bottom and SIO Sigsbee trawling and from (dark grey circles) video footage of the Kara Sea bottom in August–September 2016.**  (Map created using PanMap; *Grobe, Diepenbroek & Siems, 2003*).

cases with good visibility, an outline of submerged crabs could be seen on the surface of the muddy sediments (Fig. 7B). Although, it is possible that a few crabs were not counted due to low visibility and deep burrowing. Therefore, the crab densities calculated from the video footage can be largely underestimated.

The highest density of crabs recorded in 2016 was in Blagopoluchiya Bay (0.87 crabs/m$^2$). The crabs were small in both the Sigsbee trawl and on the video (majority of crabs with CW below 20 mm). Even though the mixture analysis identified two distinct small sized groups (8–9 mm and 14–15 mm) for both sampling methods, the trawling sample had

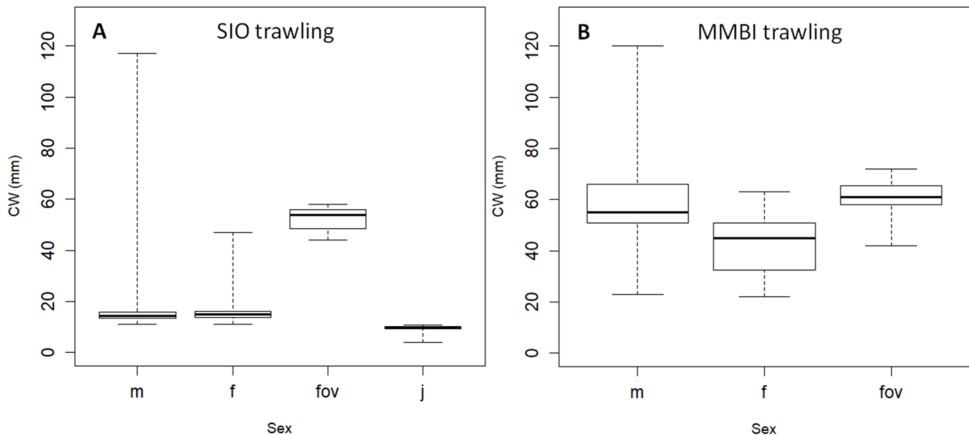

**Figure 6** Box plots of carapace width (CW) distribution of *Chionoecetes opilio* of different sexes. (A) SIO RAS, and (B) MMBI trawling samples. **m**–males; **f**–non ovigerous females; **fov** –ovigerous females; **j** – juveniles with CW less than 11 mm. Boxes reflect the data between the first and third quartile (50%), thick line is the median, and whiskers extend to the maximum and minimum CWs (see Table 4).

**Table 4** General statistics of the carapace width (CW) distribution of crabs of different sexes caught in SIO and MMBI trawling samples.

| A | | | | |
|---|---|---|---|---|
| IO trawling | Males | Females | Ovigerous females | Juveniles |
| *n* | 293 | 366 | 8 | 190 |
| min-max | 11–117 | 11–47 | 44–58 | 4–11 |
| central 50% | 14–16 | 14–16 | 49–56 | 9–10 |
| mode/median/mean | 14/15/18 | 14/15/17 | 55/54/52 | 10/10/9 |

| B | | | |
|---|---|---|---|
| MMBI trawling | Males | Females | Ovigerous females |
| *n* | 523 | 67 | 72 |
| min-max | 23–120 | 22–63 | 42–72 |
| central 50% | 51–66 | 33–51 | 58–66 |
| mode/median/mean | 52/55/58 | 48/45/42 | 62/61/61 |

**Notes.**

n, number of crabs measured; min-mas, minimum and maximum; CW, central 50%—reflect the data between first and third quartile (50%) of CW distribution (see Fig. 6)..

much sharper differences between the groups, and the larger groups were much more distinct (Figs. 3A, 3B, Table 1). Such differences could be due to possible errors in size measurements of filmed crabs. In the video samples, the CW was measured by a ruler, which has lower precision. In addition, the measurements were recalculated based on the distance between the two laser points that were also measured by a ruler. The crabs were not always in absolutely plain position towards the camera, and the visibility often did not permit to see the edges of carapace clearly. Therefore, there could be some additional noise

**Table 5** The results of the analysis of video data of Chionoecetes opilio filmed by SIO RAS in the Kara Sea in 2016.

| | Number of video frames | Total video area m² | Number crabs | Density crabs/m² |
|---|---|---|---|---|
| Blagopoluchiya Bay | 405 | 449 | 389 | 0.87 |
| Haug Bay | 578 | 629 | 7 | 0.01 |
| Tsivolka Bay | 1,077 | 1,355 | 204 | 0.15 |
| Abrosimov Bay | 629 | 1,130 | 86 | 0.08 |
| Kara Gates 2 | 443 | 213 | 118 | 0.55 |

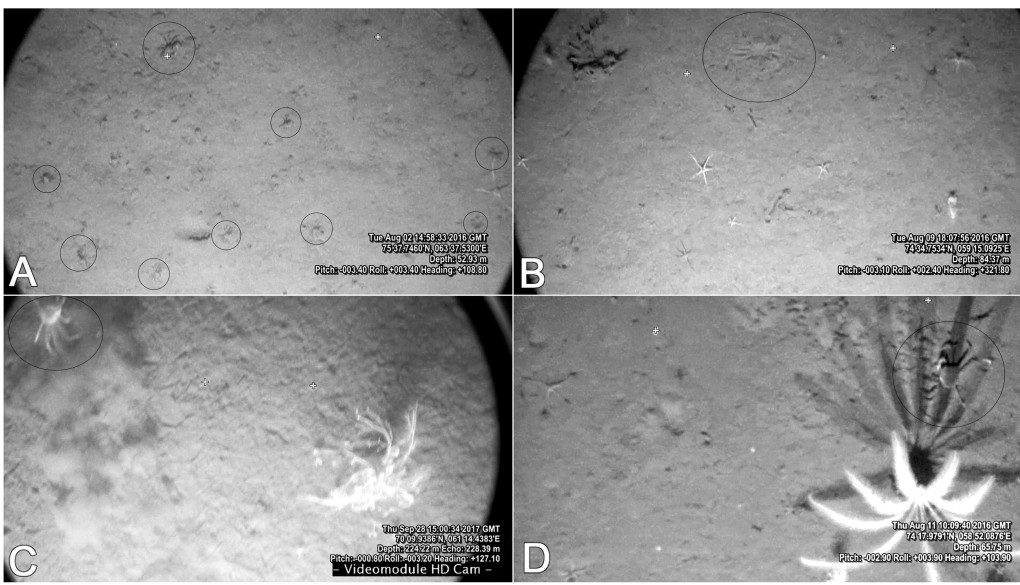

**Figure 7** **Frames from the video footage recorded by the UTSI *Video module*.** (A) Blagopoluchiya Bay frame with 8 crabs. (B) Imprint of borrowed crab on muddy sediments of Haug Bay. (C) A crab creating a sediment cloud while running away in the Kara Gates Straight (filmed in 2017). (D) Snow crab in the shadow of a sea lily in Tsivolka Bay. Crosses outline the original position of laser points 60 cm apart.

in CW measurements from the video footage in comparison to the direct measurements by calipers of live organisms.

However, the most important error in the identification of size groups using video samples was due to sexual dimorphism. The size groups (instars) of *Ch. opilio* are extensively described in the literature and the young crabs (pre puberty molting <20–30 mm) seem to have similar size groups across their range of habitat (*Ito, 1970*; *Kon, 1980*; *Sainte-Marie, Raymond & Brêthes, 1995*; *Ernst et al., 2012*). These instars are in accordance with those observed for young crabs in the Kara Sea in 2014 (*Zalota, Spiridonov & Vedenin, 2018*) and in 2016 (present study, Table 1). After puberty molting (which was shown to be at 37–40 mm for males and 17 mm for females, in the Gulf of St. Lawrence) crabs' growth rate and possible skipping of molts in females is strongly affected by temperature (*Sainte-Marie, Raymond & Brêthes, 1995*; *Alunno-Bruscia & Sainte-Marie, 1998*; *Dawe et al., 2012*), which

is less than 0 and −1 °C in most areas of the Kara shelf (*Polukhin & Zagretdinova, 2016*); see also (*Zalota, Spiridonov & Vedenin, 2018*). Further growth and survival success of larger crabs may also be affected by low food availability for benthic predators in the Kara Sea (*Zenkevich, 1963*; *Kulakov et al., 2004*). As they age further, they molt approximately once a year, or even rarer until they reach their terminal molt (males at CW (postmoult) as small as 40 mm up to 150 mm; females 30-95 mm) (*Ito, 1970*; *Robichaud, Bailey & Elner, 1989*; *Comeau et al., 1998*; *Sainte-Marie & Hazel, 1992*; *Sainte-Marie, Raymond & Brêthes, 1995*; *Alunno-Bruscia & Sainte-Marie, 1998*).

Taking into account these errors, caused by aggregating males and females in video samples, it is not surprising that crabs larger than 20 mm blur into one large size group in the video data, while crabs caught by the large MMBI trawl and measured more accurately can be separated into at least 4 size groups (Table 1, Figs. 2B, Figs. 2C). Even finer size structure in the MMBI samples can be seen if crabs over 20 mm are separated according to sex (Table 1). Males have 7 distinct size groups over 20 mm CW, whereas, females' groups are more blurred. This can be due to the differences in molting and growth rates. Since video data cannot provide information on sexual dimorphism, the work done to study exact differences in the size structure of immature and mature crabs in the Kara Sea is not presented in this paper. Nevertheless, the obtained data still permits to identify general size groups and to approximate their relative quantities.

## Development of snow crabs' invasion and its possible role in Blagopoluchiya Bay

Blagopoluchiya Bay appears to have very different size structure of the snow carb population compared to all other sampled areas. Most of the crabs, both caught in the trawl and in the video footage, were less than 20 mm, and form 2 high frequency groups at around 10 and 15 mm CW (Figs. 3A, 3B, 4; Table 1). These size groups correspond to the age of less than 2 years. In the Gulf of St. Lawrence it takes 16 to17 months for crabs to achieve CW 10 mm, and another 16 months to achieve size 20 mm through multiple molting events (*Sainte-Marie, Raymond & Brêthes, 1995*). Therefore, the majority of these crabs could not have settled much earlier than 2014. That year we caught crabs no bigger than five mm in the vicinity of that bay (station 51 in (*Zalota, Spiridonov & Vedenin, 2018*). Indeed, that was the first year when the crabs have been observed across the entire western Kara Sea and in most cases they were young, with high abundance of just settled crabs (*Zalota, Spiridonov & Vedenin, 2018*).

There were a few larger crabs in Blagopoluchiya Bay that were caught on camera. Their sizes are not big enough to assume that they actively migrated from other areas. This suggests that there were earlier successful settlings of crabs, but the proportion of larger crabs is almost negligible (Figs. 3A, 3B). The density of the young crabs in the bay is very high (0.87 crabs per m$^2$) and in some cases we observed up to 8 crabs in one video frame (approximately 0.5 m$^2$). Such high densities suggest that a recent combination of favorable oceanographic and sea ice conditions in this area facilitated a massive settling and survival of snow crabs. The larvae that settled in Blagopoluchiya Bay were likely transported by the Eastern Novaya Zemlya current.

This current originates from the Barents Sea water, entering the Kara Sea off the northern coast of the Novaya Zemlya and is directed to the south-west along the eastern coast of this archipelago (*Pavlov & Pfirman, 1995*). Previously, the bays of the Novaya Zemlya, especially in the north, such as Blagopoluchiya Bay, have been blocked by ice longer than most of the western Kara Sea (*AARI, 2007–2014*). Although, a narrow Northern Novaya Zemlya Polynya occurred from time to time (*Gavrilo & Popov, 2011*). Since 2011, changing sea conditions of the 2000–2010s manifested in an abrupt sea ice decrease in June (*NOAA, Snow and Ice, Regional Sea Ice, 1979–2018*). Although, there was more ice in the spring of 2014 (in comparison to 2011), the sea ice cover along the Novaya Zemlya Archipelago was the first to retreat and an extensive polynya was formed (*AARI, 2007–2014*). Early sea ice decay could have facilitated seasonal development of phyto- and zooplankton, and hence favorable conditions for feeding and successful settling of crab larvae.

When the first snow crabs were settling, they experienced low predator pressure in the Kara Sea benthos. Adult snow crabs that are highly cannibalistic were not yet present, and the native crab species, *Hyas araneus*, are very rare in that area (*Anisimova, Ljubin & Menis, 2007*); authors' observations). In addition, predatory demersal fishes have substantially lower diversity and abundance in the Kara than in the Barents Sea (*Dolgov et al., 2009*; *Dolgov, Benzik & Chetyrkina, 2014*; *Dolgov & Benzik, 2016*).

Initially snow crabs could have settled in the southern bays and in the central western part of the Kara Sea (in addition to larvae transportation adults could have reached it by active migration) before they reached the northern bays of the Novaya Zemlya Archipelago. Hence, we can observe such difference in the population size structure of Blagopoluchiya Bay (narrow range of size groups) in comparison to the southern areas (all size groups are present).

## Indication of formation of spatial population structure

Overall, there seems to be a pattern in the snow crabs' size distribution across the western Kara Sea. In the southern Kara Sea these patterns are not related to depth since there is no strong correlation between the CW and the depth across all MMBI stations (large bottom trawl). These stations are positioned on a broad, slightly sloping shallow shelf (46 to 195 m). However there is a visually observable trend of maximum sizes prevailing in the Kara Gates Strait and northwards along the Yamal Peninsula (Fig. 5). This closely resembles prevailing current path of the Barents Sea waters, known as the Yamal Current (*Pavlov & Pfirman, 1995*; *Zatsepin et al., 2010a*). The area along the Yamal Peninsula and the Novaya Zemlya Archipelago has higher benthic biomass rates than in the center of the western Kara Sea (*Antipova & Semenov, 1989*; *Denisenko, Rachor & Denisenko, 2003*; *Kulakov et al., 2004*; *Kozlovskiy et al., 2011*). The decrease of the maximum CW from the Yamal Peninsula towards the center of the western Kara Sea could be due to lower food availability further away from the Barents Sea influence, and thus the crabs have insufficient nutrition to achieve larger sizes. However, this hypothesis needs further confirmation.

There could also be behavioral separation, where smaller sized crabs are forced to move to a territory with less food to escape cannibalism, which is very common among this species (*Conan et al., 1996*; *Comeau et al., 1998*). No such trends can be observed along the

Novaya Zemlya Archipelago, probably due to low sampling effort. There are reports of difference in the habitat preferences of different sized snow crabs in their native habitat areas (*Comeau et al., 1998*; *Ernst et al., 2012*). The bays of the Novaya Zemlya have the potential to act as a nursery for smaller, more vulnerable specimens, as it has been observed in the Gulf of St. Laurence (*Comeau et al., 1998*). Whether this separation exists or will ever exist in the Kara Sea is hard to say at this point. However, the vicinity of deep trough along the Novaya Zemlya Archipelago could attract larger crabs and lead to size related migration out of the bays.

The recruitment of crabs at the early stages of population establishing in the Kara Sea might be mostly due to the inflow of the larvae from the Barents Sea (*Zalota, Spiridonov & Vedenin, 2018*). Here we present findings of substantial number of ovigerous females, most of which have been found along the Yamal Peninsula (with no apparent spatial or depth distribution patterns), and none in the bays. It is hard to say whether this was due to sampling gear limitation to catch representative sample of larger crabs or a reflection of the real picture. All ovigerous females had CW larger than 40 mm (Fig. 6). This corresponds to the size of female's terminal (sexual maturity) molt reported in the literature (starting from 35 mm) (*Sainte-Marie, Raymond & Brêthes, 1995*; *Alunno-Bruscia & Sainte-Marie, 1998*). Crabs of these sizes had a low catchment rate in the SIO trawling samples along the Novaya Zemlya Archipelago. In most cases the video samples suggest that crabs with CW larger than 40 mm prevail in the vicinity of most sampled bays (Fig. 4). Therefore, it is likely that reproducing crabs are present in most sampled areas. It is safe to say, that at present the snow crab has a reproducing population in the Kara Sea.

The spatial structure of the snow crab population in the Kara Sea is still in process of formation. The data of 2016 indicate that this process may lead to a quite complex system, which is based on local recruitment, transport of larvae from the Barents Sea and across the western Kara shelf, formation of nursery grounds, and an active migration of adults and their concentration in particular shelf areas with appropriate feeding conditions. This system on the other hand can't be static as it is influenced by changing advection of the Barents Sea water and its interaction with the water of river discharge origin (*Zatsepin et al., 2010b*), sea ice regime, trophic conditions and predation pressure on juvenile crabs.

## CONCLUSION

The present study compares and combines the results obtained using three different sampling gear to assess the size composition and density of the snow crab population. Smaller Sigsbee trawl allows catching of small crabs, even those that are just settled. A large commercial type trawl catches large agile crabs and results in a larger number of ovigerous females. Video transects probably underestimates smaller crabs, but gives a rapid and accurate estimate of larger crabs' densities. This method is helpful to monitor the spatial progress of the crabs' invasion and the appearance of commercial sized crabs. Trawling is necessary to study reproductive biology of crabs in new conditions and to carefully identify size structure of the population. The data of 2016 has finally proven that the Kara Sea snow crab population is reproducing, although presumably still strongly influenced by the

larval transport from the Barents Sea. We observed initial nursery areas in the bays of the eastern coast of the Novaya Zemlya Archipelago. A number of commercial sized crabs can be observed near the Kara Gate Strait and along the western coast of the Yamal Peninsula with higher food availability than the rest of the Kara Sea.

## ACKNOWLEDGEMENTS

We would like to thank the Captains, the crews and the science teams and their leaders that participated during the RV *Akademik Mstislav Keldysh* SIO and RV *Dalniye Zelentsy* MMBI cruises to the Kara Sea in 2016. We are particularly grateful to the team of engineering and technical research of SIO for creating and using an uninhabited, towed, submerged, inert vehicle (UTSI) *Video Module* to film the footage during the SIO expedition.

### Funding

The research was supported by the Russian Foundation for Basic Research (grant No. 18-05-70114). The funders had no role in study design, data collection and analysis, decision to publish, or preparation of the manuscript.

### Grant Disclosures

The following grant information was disclosed by the authors:
Russian Foundation for Basic Research: 18-05-70114.

### Competing Interests

The authors declare there are no competing interests.

### Author Contributions

- Anna K. Zalota and Olga L. Zimina analyzed the data, contributed reagents/materials/analysis tools, prepared figures and/or tables, authored or reviewed drafts of the paper, approved the final draft.
- Vassily A. Spiridonov analyzed the data, prepared figures and/or tables, authored or reviewed drafts of the paper, approved the final draft.

### Data Availability

The raw data is available as a Supplemental File.

### Supplemental Information

Supplemental information for this article can be found online at http://dx.doi.org/10.7717/peerj.7952#supplemental-information.

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
