# Peer review of "Combining data from different sampling methods to study the development of an alien crab Chionoecetes opilio invasion in the remote and pristine Arctic Kara Sea"

_PeerJ, doi:10.7717/peerj.7952_

## Round 0.1 · original submission · Minor Revisions

Two reviewers have now read your manuscript, and both agree that it requires minor revisions, a decision with which I agree. Note that one reviewer suggests that you could change your focus and make your aims wider. I see their point, although this decision should remain yours.

I would like you to reconsider your use of the word 'invasion'. Although it is an appropriate verb, the word "invasive" has a formal meaning in invasion biology. Your crabs do seem to fit this, but you do not formally state that this species fits the definition of an invasive species. This is confused by you stating that it is "less invasive" (L132). Many authors have defined terms in invasion biology, although the generally accepted concepts are to be found in Blackburn et al. 2014 (Blackburn, T.M., Essl, F., Evans, T., Hulme, P.E., Jeschke, J.M., Kühn, I., Kumschick, S., Marková, Z., Mrugała, A., Nentwig, W. and Pergl, J., 2014. A unified classification of alien species based on the magnitude of their environmental impacts. PLoS biology, 12(5), p.e1001850).

[]

·

Basic reporting

Text is written in professional English, although some minor stylistic corrections are needed.

The paper is well structured. Some text should be narrowed down to Tables.

In some places text looks redundant and may be easily shortened (more comments provided in attached file)

Experimental design

no comment

Validity of the findings

Discussion may be better structured and shortened (some parts can be moved to Materials and Methods)

Additional comments

This is an interesting and useful paper. It requires some revision - more details are given in attached file

·

Basic reporting

Basic reporting

Professional editing and cleaning of the language are necessary throughout the document before the publishing. Please foresee that e.g. the sentence (line 297-298) “Each video frame was visually inspected by A. Zalota and only frames with tolerable visibility were used” -> where both the person and the range of the “tolerance” is unknown and need explanations. Such “undefined” wordings and sentences need to be cleaned up throughout the document, and I have given some few (but not all) examples in the document.

Intro & background are showing the context well, but need to be divided into appropriate subchapters. Please see comments throughout the documents about moving the text to appropriate sections.

Please avoid using Institution names, Cruises, and Person names (see e.g. line 50-58) outside the “Material and Method” section, but only use appropriate references to what has been published from these Institutions, Cruises, and Persons.

Please use a consistent name of the snow crab. Several names are used throughout the manuscript as it is now.

I have not had the time to look through the Literature and referenced list, but have added some comments on the document on relevance.

The figure text in figure 3 and 4 are very similar .. could these two figures be merged?

Time limits prevented me from considering the Raw data supplied.

Experimental design

EXPERIMENTAL DESIGN

Time limitation made it difficult to carefully read the “Original primary research within the Scope of the journal” and evaluate if this publication falls within this scope. I hope the editor of the Journal can help here.

The research question is not well defined at the end of the Introduction. I have suggested a way to do this that I find relevant & meaningful. But it is up to the auditors to evaluate what they will like.

The research fills an identified knowledge gap: Because the snow crab only recently has been recorded in the Kara Sea (i.e. a new spreading species), this work is novel, important and timely.

Rigorous investigation performed to a high technical & ethical standard: The authors use three different sampling methods and draw conclusions across these methods. The conclusions are that the Sigsbee trawl with small mesh-size and Video sampling obtain information of both large and small crabs, while the large bottom trawls only on large crabs because of a large mesh-size. Trawl sampled individuals can be accurately measured, while Video sampled individuals only occasionally. The combined use of these gears is recommended by the auditors to provide a full data-scope of the existing size groups in a population.

I find that the 3 methods are described with sufficient detail & information to replicate.

Validity of the findings

VALIDITY OF THE FINDINGS

Knowing the propagule areas, the reproductive and spreading potential of the Kara Sea snow crab population will have an Impact on the forecast of the future of the Siberian Shelf ecosystems, and the options for resource management and biodiversity conservation.

Because few studies are made on the Kare Sea snow crab population due to the remoteness of this area. Combined with the fact that the spreading of the crab is ongoing, makes this paper novel.

Data are robust as far as possible, statistically sound, & controlled. But as the authors state: Three different methods are combined and evaluated as far as possible. No standardized sampling system exists in the remote Kara Sea, and this work has made a good attempt to compile all information from these different methods.

The Conclusion paragraph should be better stated, linked to the research question (that need to be clearly written in the end of the Introduction” & limited to supporting results.

Additional comments

Combining data from different sampling methods to study the development of an alien crab Chionoecetes opilio invasion in the remote and pristine Arctic Kara Sea, by Anna K Zalota Olga L Zimina Vassily A Spiridonov

Review based on Dr. Lis Lindal Jørgensen:

This work is a description of the spatial distribution of snow crabs in the Kara Sea, mainly based on the size and sex of the crab. Because the snow crab only recently has been recorded in the Kara Sea (i.e. a new spreading species), this work is novel, important and timely and should be published. As the authors correctly and timely states: it is critical to study the development of the snow crab population in the Kara Sea to forecast the future of the Siberian Shelf ecosystems, and the options for resource management and biodiversity conservation.

The authors use three different sampling methods and draw conclusions across these methods. The conclusions are that the Sigsbee trawl with small mesh-size and Video sampling obtain information of both large and small crabs, while the large bottom trawls only on large crabs because of a large mesh-size. Trawl sampled individuals can be accurately measured, while Video sampled individuals only occasionally. The combined use of these gears is recommended by the auditors to provide a full data-scope of the existing size groups in a population.

My opinion is, that while the title: “Combining data from different sampling methods to study the development of an alien crab Chionoecetes opilio invasion in the remote and pristine Arctic Kara Sea” and the paper is focusing on “sampling methodology”, I suggest turning the focus to study: “The development of an alien crab Chionoecetes opilio invasion in the remote and pristine Arctic Kara Sea” and therefore focus on 3 goals (see also line 138-140):

We will like to 1) identify from where the snow crab has entered the Kara Sea by population size, 2) identify brooding areas in the Kara Sea by measuring size distribution of snow crab populations, 3) to evaluate if the snow crab population in the Kara Sea is self-sustained by measuring the population size of mature, egg brooding females.

This should be clearly expressed in both the abstract, discussion and in the conclusion.

---

## Round 0.2 · Minor Revisions

In my previous decision, I stressed the need for you to use stipulated invasion biology terms. However, in your rebuttal I note that you suggest trying to remove the term 'invasive', which is a misunderstanding of my instruction.

Currently, the manuscript provides conflicting statements including the first sentence: “…invasion and naturalization can take a very long time.” as naturalization is the precursor of invasion, this sentence is not logical. Moreover, this same concept of such lag phases is more commonly referred to as “invasion debt” and has a set of new literature associated with it. Other places would benefit from a defined framework (e.g. L499). Using defined terms and citing the relevant literature is an important part of communicating your study to a wider audience, and this is especially important in a general journal like PeerJ. Similarly, because ‘invasive’ is a defined term, please avoid using it in another sense (L150).

Secondly, you state that the English has been corrected, but that you are sure that there are “a lot of small mistakes”. I agree that there are still many mistakes and I cannot accept your manuscript until they are all corrected. It appears that your own skills and those of your native speaker are not enough to correct this manuscript. I suggest that you need to find better help, or use a professional service. Note that although I have highlighted many instances in the introduction, these were so numerous that I do not have time to make a thorough sweep of the entire document. This remains your responsibility as authors, and cannot be passed onto reviewers or editors.

In addition, I was concerned about the statement in L217: This appears to suggest that you have removed some data subjectively. If this was done methodologically, you should give the proportion of the data removed. Moreover, it’s not at all clear why you mean by ‘cutting frames around the edges’. This needs a coherent explanation.

I look forward to your revision.

---

## Round 0.3 · accepted · Accept

Thank you for your revision. Please note that although I've chosen to accept your manuscript, there are a large number of grammatical errors that were made in the most recent changes. I've highlighted these in the attached text. Make sure that you make the necessary requirements at the next stage of production.